# Social Distancing Impact on Higher Education during COVID-19 Lockdown

**Ionel N. Sava** 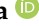

Department of Sociology, University of Bucharest, 050663 Bucharest, Romania; nicu.sava@unibuc.ro

**Abstract:** During the COVID-19 pandemic, education online meant limited or no in-person interaction with professors and peers. In this article, research questions look for social distancing impact on subjects attending computer-mediated education. Educational technology factors were selected and exposed to students' evaluation in a semi-structured questionnaire. Results confirm that online education increased students' acceptance and positive attitude towards digital learning for 8 out of 10 subjects. On the other hand, factors that drive motivation showed diminished satisfaction with content for 4 out of 10 students and reduced capacity to stay focused for 7 out of 10 students. This research points toward factors that convert interaction with peers and instructors to such an extent that they impact basic educational fields such as motivation and satisfaction. There were interrogated social interactivity factors, as half of the subjects reported missing learner–learner and learner–instructor interaction. Results showed that up to one third of surveyed students showed diminished motivation alongside less satisfaction with content. The article concludes that digital education should multiply and adapt its own content and delivery routines and it suggests that the online education experience should serve development of computer-mediated learning as well augmenting of in-person education.

**Keywords:** COVID-19; social distancing; online teaching; interactivity; higher education

## 1. Introduction

In the spring of 2020, at the outbreak of the COVID-19 pandemic, most of the European Union countries were introducing new education technologies. This was part of the Bologna education reform and it was intended to increase the share of what had previously been called remote education, distance learning, or, more recently, online education. In both public and private universities, a number of courses and programs benefitted from online support. However, basic higher education programs still required class attendance. As such, crowded amphitheaters where eloquent professors delivered their courses had survived education reforms. In the end, no-one thought the pandemic would send all of them online for the next four semesters and make them adapt to the digital format sooner than expected.

By studying first-hand reactions at the end of the online education period, this article explores opportunities and challenges students faced during the closure of the in-person learning. This research topic is instrumental for adapting higher education to the conditions of globalization of pandemics and climate change. Using technology to improve education is a goal that aids the sustainability of both natural and social environments, and even more so for developing countries.

Educational technology research has intended to identify factors that influence attitudes, intentions, and behaviors and which are measurable using available techniques. Most of them are inspired by existing evaluation procedures. Lately, a sort of semantic alignment of the identified constructs has allowed these techniques and procedures to be grouped according to measurement intent (Kemp et al. 2019). The available literature eventually reveals a diversity of constructs and competing methods of measurement that need to be integrated. For instance, commonly used external factors were integrated by

Abdullah and Ward (2016) and this resulted in a General Extended Technology Acceptance Model for E-Learning (GETAMEL). This model selected the five most-used factors that influence online education.

Drawing on a survey of 132 college students, Doleck et al. (2018) evaluated GETAMEL and validated it from a quantitative point of view by employing a partial least square path modeling approach—i.e., estimation of complex cause–effect relationship—which is accurate most of the time. However, a number of situational factors influencing determinants of e-learning acceptance are suited for qualitative approach as well.

Such inadvertencies led Kemp, Palmer, and Strelan to argue that "it is important that measurement models cover an inclusive scope and measure all likely factors in a way that brings consistency from study to study" (Kemp et al. 2019, p. 2397). Therefore, all factors shown or theorized to be influential were simplified and incorporated in an organized collection of primary, secondary, and tertiary taxonomy groups to contain as many as 61 measurement constructs.

Alongside factors affecting attitudes, intentions, and behaviors, this taxonomy includes "social interactivity" factors (part of the instructional attributes group) which are measured as learner–learner and learner–instructor interaction, as well as learning group cooperation and competition. Therefore, for this research, I selected Kemp, Palmer, and Strelan's taxonomy, taking into account its theoretical relevance and methodological utility. The survey was taken at the end of two years of online education experience.

A. Patricia Aguilera-Hermida surveyed 246 students a few weeks after the start of the COVID-19 lockdown using this taxonomy. The results support the idea that "online or remote education implies that students are physically distant from the instructors and require a delivery method" while "many students around the world had to transfer from face-to-face instruction to an online learning environment in the middle of the semester" (Aguilera-Hermida 2020, p. 1). Interaction between teachers and students mediated by technology is quite different to in-person interaction and it does have significant influence on the educational outcomes. Among other things, it means that "if students lack confidence in the technology they are using or do not feel a sense of cognitive engagement and social connection, the result may affect negatively the students' learning outcomes" (Aguilera-Hermida 2020, p. 2).

The research hypothesis of this article is that social distancing during lockdowns increases students' expectation and positive attitude towards online education technology, yet it converts interaction with peers and instructors to such an extent that it impacts basic educational factors, for instance, motivation and satisfaction with content.

In order to check this hypothesis, the article addresses three research questions. The first one refers to the relevant factors that measure online education and how students rate them in comparison with in-person education. The second question refers to factors exposed to social distancing influence and how to measure their exposure. The third question asks what the impact of social interaction switch is on higher education in general.

The selected subjects for this research were exposed to both online and in-person education. The results are preliminary and the interpretation is limited to a sample of students randomly extracted from particular universities. Further research should be pursued in larger educational contexts in order to advance consolidated conclusions and recommendations.

For the time being, it is already common sense that online education and in-person education are complementary methods of delivering knowledge, while their associated pedagogies are rather different, as are their outcomes.

Considering students' perceptions and attitudes, universities could better and faster develop education programs that increase their digital content. A good number of studies suggest there are certain benefits, while others warn that the negative impact is not yet fully evaluated. What is clear for now is that responding to disruptive times means investing more in programs and pedagogies that require alternative methods to the face-to-face

teaching, while resources could be directed to develop digital strategies (Purcell and Lumbreras 2021).

In a reflection paper on the future of digital and online higher education in Europe, the European Commission mentioned that nearly 90 percent of the universities in the European Higher Education Area have a strategy for digitally enhanced learning and teaching (Humpl and Andersen 2022). However, this article argues that online experience is also relevant to redesign the role of in-person education that is still at the core of higher education in general.

## 2. Research Design and Method

### 2.1. Selecting the Factors

Specific to online pedagogy is the elusive control over the education process. Therefore, effective pedagogies have continued to develop for a few decades now and sociologists look for what is called flipping the classroom (Forsey et al. 2013). The main issue was described as early as the 1980s and pointed out that the influence that goes along with communication technologies is exposed to degradation, errors, and omissions. On the one hand, with online communication, "noise" occurs and, hence, influence and intensity of messages degrade when transferred by technical means, that is to say, algorithms can perform many things except replicate human interaction. On the other hand, there is a sort of "channel capacity", a human mental ability of an audience (otherwise called "valence") to adequately receive and reply or convey information (Friedman 1980). For such obvious reasons, any measurement of online education should estimate both technical and human capacity to perform specific educational tasks.

Universities that were experiencing web-based interactive learning using audience response systems (ARSs), a sort of pop-in technical system which allows students to react directly during online class, reported rather good results (McKenzie and Ziemann 2020). Pupils had the opportunity to react in a timely manner and have their contribution be properly measured. Odeh et al. presented the Poll Everywhere Audience Response System experience during the COVID-19 lockdown for 140 medical school students. The Poll Everywhere platform performed rather well as "[it] was clearly supported by the fact that 59.4% of students strongly agreed and 32.1% agreed" to a certain degree (Odeh et al. 2022, p. 104).

Research that looks for factors affecting attitudes towards educational technologies has been only recently systematized. In the universities that were using online teaching but not interactive platforms as well, teachers addressed live cameras and microphones without acknowledging how many persons online were altogether "there". Online lectures and seminars were the prevailing methods engaged during the COVID-19 lockdown.

As mentioned already, more objective measurement asks for factors to be grouped into taxonomic groups and to be coupled with appropriate behavioral theories underpinned with technology acceptance and technology use models (Kemp et al. 2019).

Taking into account that smart phone applications are also used, a similar taxonomy is worth mentioning for mobile devices as well (Nickerson et al. 2009).

I selected six factors to be discussed in this article. They are (a) technology acceptance, (b) attitudes, perception, and motivation, (c) perceived behavioral control, (d) cognitive engagement, (e) social factors (social norms and influence), and (f) social interactivity (as part of instructional attributes). These factors are selected from the Kemp et al. (2019) model and they are used here as measurement constructs.

### 2.1.1. Technology Acceptance

Technology acceptance implies "the willingness and the continuous use of technology from the user" (Aguilera-Hermida 2020, p. 2). It was developed by the theories of reasoned action and of planned behavior. It means, for instance, "the person's perception of the social pressures put on him or her to perform the behavior" (Ajzen and Martin 1980, p. 6). It is currently used by the technology acceptance model (TAM) and it is defined as "the

user's subjective probability that using a specific application system will increase his or her job performance within an organizational context" (Davis et al. 1989, p. 985).

Due to the fact that emergency decisions offered no time to devise a properly designed techno-tool, universities opted for already existing software that was tested by trial and error. Professors and students had no option but to take it or leave it; soon after, this turned into liking it or not. That generated a sort of polarization with regard to technology acceptance. Therefore, available education software such as MSTeams, Google Meet, and Zoom were amongst the most used during the pandemic in the universities the subjects of this research were selected from. In due time, universities created their own software or adapted the existing ones to their academic identity. On the other hand, most students in this research used multiple devices to access online education. Mobility and accessibility increased the probability of technology acceptance during the COVID-19 lockdown. Technology acceptance measurement eventually reveals the digital profile of a generation born at the turn of the millennia.

### 2.1.2. Attitudes, Perception, Motivation

Attitudes towards use refer to "the individual's positive or negative evaluation of performing the behavior" (Ajzen and Martin 1980, p. 6). Affect or perception includes user enjoyment and satisfaction with the operation of the information system, the education platform, and the subject's emotional condition. Affects are "liking for particular behaviors" (Compeau and Higgins 1995, p. 196). Good learning outcomes are usually associated with positive attitudes and affectionate emotions. In this research, a number of items address both enjoyment and displeasure of online education with its ups and downs during the COVID-19 pandemic.

Motivation is driving the subject's determination to learn, that is, students' wishes to perform. Therefore, motivation is about subject's perceived relevance or utility of a given activity that drives the behavioral intention. Humans perform something to the extent there is a utility to be achieved, and for that there is a prior motivation. Motivation means "performing a behavior in order to achieve some separable goal" or behavior "performed for itself, in order to experience pleasure and satisfaction inherent in the activity" (Vallerand 1997, p. 271).

The lack of motivation makes online education less attractive for students and increases the time they need to fulfill course requirements. Motivation is brought about by cooperation and competition which are defining characteristic of groups of peers (schooling class) in general.

Hence, what is real cognitive process in online education is about to be determined.

### 2.1.3. Perceived Behavioral Control

As described by Kemp et al. (2019), perceived behavioral control refers to the ease of use (difficulties encountered) and complexity (as perceived by subjects), as well environmental and situational contexts, such as facilitating conditions for using technology and having access to distance learning at all. Self-efficacy comprises the subject's judgments of his or her own capacity to attain specific tasks that peers usually perform. It is about "peoples' judgements of their capabilities to organize and execute courses of action required to attain designated types of performances. It is concerned not with the skills one has but with judgements of what one can do with whatever skills one possesses" (Bandura 1986, p. 391). It is an individual's estimate of what can be achieved with the knowledge and skill under his or her control. Self-efficacy is part of the social cognitive theory developed by the 1970s. A. Bandura described it as "a person's estimate that a given behavior will lead to certain outcomes" (Bandura 1977, p. 193, quoted by Kemp et al. 2019, p. 2400). Currently, social cognitive theory considers individuals as self-motivating agents in the engagement of their own actions, emotions, cognition, and motivation. In e-learning, self-efficacy is "the degree of ease with which a university student can access and use a campus e-learning system as an organizational factor" (Park 2009, p. 153). It means that skills are already

achieved when learning software is to be used for online education. One could compare self-efficacy with bike-riding. Once skills are achieved, one could use any kind of digital device with minimum adjustment effort.

Accessibility is a factor in the perceived behavioral control class and resembles the degree to which the students have access to reliable Internet outlets in the conditions of a reasonable degree of mobility (no restrictions of time and place). The extensive use of smart phones by the young generation leads us to include accessibility and mobility among the factors influencing online education. Android and IOS already have educational applications, and universities should formally include them among the available pedagogic techno-tools.

### 2.1.4. Cognitive Engagement

Social psychologists refer to cognitive engagement as absorption, a sort of human mental sponge that uses information flow and concentration to perform cognitive absorption. Cognitive engagement requires attention, focusing on essentials, looking for details, and concentration for a specific period of time. (Kemp et al. 2019, p. 2407). The flow of information should be properly proportioned in order to fit the receiver's capacities. Usually, adapting information flow to students' average capacity gives them the opportunity for cognitive engagement. (Cheng 2013) Specific to the COVID-19 pandemic's emergency switch to online education was that students had to adapt to the loose-fitting flow of information, usually exceeding their capacity to engage. This research addresses this factor using specifically designed questions.

### 2.1.5. Social Factors

The social polarization during the COVID-19 lockdowns extended to education, as both parents and pupils spend most of the time at home. Successive lockdowns and a range of health restrictions were in place for as long as two years. Stress and anxiety were eventually reported. Therefore, they were also measured in this research in order to test their influence on such educational factors as self-efficacy (trust in own abilities to perform tasks), perceived relevance (motivation), and affect (satisfaction) with education-related activities. They are the situational factors that have a certain impact on education in general. Triandis describes them as "the individual's internalization of the reference groups' subjective culture, and specific interpersonal agreements that the individual has made with others, in specific social situations" (Triandis 1980, p. 210). Facilitating conditions refers to "objective factors . . . that . . . make an act easy to do" (Triandis 1980, p. 205; quoted by Kemp et al. 2019, p. 2400) and include physical and emotional conditions at home, digital accessibility (sharing computer and room with siblings, Internet outlet), and the like.

### 2.1.6. Social Interactivity

The substitute for the face-to-face interaction is the so-called "connective action" that is computer-mediated interaction, which is rather difficult to capture in numbers (Bennett and Segerberg 2012). Social distancing changed the learner-to-learner and learner-to-teacher interaction, while cooperation and competition of peers took a digital turn. This research looked for students' perception of missing colleagues, impersonal teachings, and lack of human (face-to-face) interaction, feedback from professors, and informal online class activities. Additional online group activities were searched for as students currently used social media groups, WhatsApp, or Instagram to diminish social distancing influence. Worth noting here is that the lengthy and lively informal class discussion outside the formal channel was better for the individuals and group cohesion.

Socializing and competing with peers seem to play important roles in learning indeed. Inter-*action* requires communication and competition of peers as guided by the instructor in a social arena such as the classroom for which the online arena is but a substitute. Having this meaning, "instructor–learner interaction ("the degree of online interaction between instructors and learners via the e-learning system") and learner–learner interaction ["the

degree of online interaction between learners and other learners via the e-learning system" (Cheng 2013, p. 75)] rely on the ability of the technology to enable these forms of social interaction important to learning" (Kemp et al. 2019, p. 2407).

Connective action—computer-mediated collective behavior—does have motivational potential, and the results here and elsewhere encourage us to search for it.

## 2.2. Selecting Subjects

A total of 114 participants were asked to give consent to and answer a semi-structured questionnaire. The survey was administered during the first week of the spring 2022 semester and was geared at collecting students' perceptions, attitudes, activities, and emotions, as well as lessons they learned during the pandemic. From an opportunity point of view, the period coincided with the return to in-person education.

In the end, 100 students were selected from the Universities of Bucharest (UNIBUC) and Timisoara (UVT) that completed the survey and had at least two semesters' online learning experience and one semester in-person attendance. The University of Bucharest sub-sample of 50 students comprises ten postgraduate students who experienced extensive learning in both online and in-person education. The two sub-samples are demographically balanced regarding age and gender. Yet, there is a sixty-five per cent female ascendancy in the aggregated sample that eventually mirrors Romania's humanistic and social sciences higher education gender balance in general. Due to a presumed cultural homogeneity, participants were not asked about ethnic identities. They received no payment or other incentive and there is no apparent bias to be reported. The answers were anonymously provided.

From a quantitative point of view, the questionnaire comprises demographic data (age and gender) as well a number of items intended to measure attitude, motivation, and engagement towards online education.

## 2.3. Methods

This research used a similar approach to Aguilera-Hermida (2020) that looked for students' perception with the swap to online education. A semi-structured questionnaire with 27 enquiries was distributed to the selected subjects. There is a difference in timing as this research was engaged after the students returned to in-person education and, taking into account the polarization premises, there is a difference in measurement methodology, as a three-point Likert scale design was mostly used for this study. Other things being equal, a three-point scale offers the polar points such as agree–disagree and like–dislike, while it lessens the neutral position. Subsequently, a Pearson chi-square test of independence was performed to check for answers' mutual exclusiveness.

The aggregated sample is constructed for one hundred subjects. A number of factors are listed—acceptance of technology, attitude toward use, motivation, and facilitating conditions—alongside distribution of preferences that reveal the influence of each of them in the online schooling evaluation process. Table 1 presents the distribution of preferences (frequencies) for the sub-samples and the aggregate sample (n), the number of valid answers, and the chi-square ($\chi^2$) test value for as many as eight educational factors. Table 2 shows the distribution of perceived abilities to perform online education tasks under stress and the chi-square test value for a second panel of situational factors. Therefore, worries about the pandemic, the individual level of anxiety, and intention to abandon schooling were measured as well. A number of cognitive engagement factors (class assignment, feedbacks from professors, exams, and grades) are also rated. Table 3 mirrors a number of qualitative categories that resulted from the open questions the participants answered and the corresponding frequencies with which they were received.

**Table 1.** Distribution of preferences (frequencies).

| Frequencies by Sample Item | Like/Neutral/Dislike | | | $\chi^2$ * | *n* |
|---|---|---|---|---|---|
| | UNIBUC | UVT | Aggregate | | |
| Technology acceptance | 30/16/4 | 18/19/13 | 48/35/17 | 14.53 | 100 |
| Attitude toward use | 34/1/15 | 26/0/23 | 60/1/38 | 53.87 | 99 |
| Affect (satisfied with content) | 3/31/15 | 2/24/24 | 5/55/38 | 39.58 | 98 |
| Motivation | 16/22/9 | 27/20/2 | 43/42/11 | 20.68 | 96 |
| Difficulty | 13/7/21 | 16/26/7 | 65/0/32 | 65.34 | 97 |
| Facilitating conditions | 34/3/13 | 35/3/12 | 69/6/25 | 62.67 | 100 |
| Accessibility | 42/8/0 | 38/12/0 | 80/20/0 | 99.08 | 100 |
| Cognitive engagement/focus | 5/13/32 | 2/12/36 | 7/25/68 | 59.95 | 100 |

* The chi-square test significance level is $\alpha = 0.05$ and the critical value is $\chi^2 = 5.99$.

**Table 2.** Distribution of perceived abilities to perform online education tasks.

| Frequency by Sample | More/Same as Before/Less | | | $\chi^2$ * | *n* |
|---|---|---|---|---|---|
| | UNIBUC | UVT | Aggregate | | |
| *Self-efficacy under stress* Item | | | | | |
| Worried about pandemics | 38/1/10 | 32/0/12 | 70/1/22 | 22.64 | 93 |
| Anxiety/Stress | 25/4/20 | 24/5/21 | 49/9/41 | 27.15 | 99 |
| Missing most (colleagues) | 34/11/5 | 30/1/19 | 64/12/24 | 44.48 | 100 |
| Time-consuming | 42/2/1 | 46/1/0 | 88/3/1 | 160,88 | 92 |
| Abandon studies | 5/25/20 | 12/4/31 | 17/29/51 | 18.39 | 97 |
| *Cognitive engagement scaling* Item | | | | | |
| Homework | 25/16/9 | 23/16/11 | 48/32/20 | 11.84 | 100 |
| Class assignments | 19/26/5 | 16/26/7 | 35/52/12 | 24.42 | 99 |
| Feedback from professors | 15/25/10 | 5/41/1 | 20/66/11 | 53.84 | 97 |
| Exams | 13/7/21 | 15/4/21 | 28/11/42 | 17.85 | 81 |
| Grades | 13/7/21 | 16/10/21 | 29/17/42 | 10,65 | 88 |

* The chi-square test significance level is $\alpha = 0.05$ and the critical value is $\chi^2 = 5.99$.

**Table 3.** Qualitative data display.

| Field | Category | | Students (*n*) | | |
|---|---|---|---|---|---|
| | | | UNIBUC | UVT | Total |
| Challenges | Circumstantial | Missing friends and colleagues | 24 | 30 | 54 |
| | | Worried about pandemic | 37 | 32 | 69 |
| | | Missing open air activities | 11 | 20 | 31 |
| | Educational | Too much time online | 22 | 16 | 38 |
| | | Increased assignments | 18 | 24 | 42 |
| | | Too-busy schedule | 23 | 28 | 51 |
| | | Missing study trips/internships | 8 | 16 | 24 |
| | | Computer skill differences | 11 | 8 | 19 |
| | Emotional | Difficulties focusing | 14 | 18 | 32 |
| | | Human interaction | 23 | 27 | 50 |
| | | Impersonal teaching | 6 | 9 | 11 |
| Positive | Safety and family | Supported by family | 40 | 42 | 82 |
| | | Comfortable and safer at home | 28 | 32 | 60 |
| | | Protected against COVID-19 virus | 21 | 18 | 39 |
| | Personal improvement | Developed computer skills | 20 | 26 | 46 |
| | | Good feedback from professors | 9 | 6 | 15 |
| | | Take care of job too | 17 | 12 | 29 |
| | | Keep in touch on social media | 22 | 12 | 34 |
| | New learning | Visual content increased | 14 | 12 | 26 |
| | | Classes more interactive | 6 | 8 | 14 |

From a qualitative point of view, a number of questions are open-ended by design, with the intent to let the subject describe their personal experience with online schooling. Reactions to the restrictions and limitations imposed during the pandemics are an important group of answers and are to be briefly mentioned here as well. A number of creative feedbacks resulted and they were coded and grouped in themes (categories) that relate to factors or items which are relevant for this research.

## 3. Previous Research

Universities operating in similar social and cultural contexts report findings close to this research. Looking for evidence that supports transition to online education, a research team from the Bucharest University of Economic Studies and the Romanian–American University focused on the intensive-oriented factors. As such, there was computed instructor activity, continuous interaction, student self-confidence, and course structure, as well as technology and administrative support (Edu et al. 2021).

Comparable results are also offered by Roman and Plopeanu (University of Iaşi) with a case study on higher education in economics. It proved that situational factors (Internet access, lack of time, inadequate working space at home) seriously impede the effectiveness of the online learning (Roman and Plopeanu 2021). Poor conditions at home recommend education in schools. Moreover, they confirmed that, ceteris paribus, the psychological distress and strain under the pressure of lockdown (social distancing) had a negative effect on online effectiveness in general.

## 4. Results

### 4.1. Technology Acceptance

In most cases, acceptance receives two possible answers (yes and no). As for the attitude towards the use of online technology, the participants rated a number of factors on a scale where 3 = like, 2 = neutral, and 1 = dislike. A complementary open question used the same attitude in order to test for consistency. For most of the answers here, no systematic discrepancy occurred. It is worth mentioning that the survey collected matured attitudes soon after the lifting of the health restrictions. Online education acceptance exposed the feeling that it was the contingency rather than the opportunity that informed the university's decision to go online.

Distribution of preferences indicates that the selected factors pass the independence test. In most cases, observed distribution is quite different than the expected distribution. Clear polarization is recorded for relevant factors (such as difficulty in using education technology) while high preferences for neutral position drive some factors (such as acceptance) closer to the chi-square test's critical value.

On the one hand, technology acceptance seems to be associated with the learning experience, as it is higher when schooling experience is lower. It also seems that digital generations take schooling similar to a new smart phone application which already integrates content specific to social media in general. Emoticons, for instance, are extensively used by today's education platforms. On the other hand, online education distributes content which is accessible but not necessarily insightful. This research checked for technology acceptance as students and teachers presumed that online education is a challenge to be taken on. After this experience, it is worth discovering to what degree digital education should be informed by traditional in-person education, or not. Self-efficacy factors suggest that confidence in technology use is not associated with in-person education, that is to say, digital education develops its own content and delivery methods.

### 4.2. Attitudes, Perception, Motivation

Participants were asked to concentrate on the online schooling and to point out what they like or dislike most. Attitude toward use (TAM) checked an individual's positive, neutral, or negative viewpoint. A number of factors, such as the feedback they had from professors, the computer-intermediary relationship with classmates, new things

they learned about, and the exams' online perception, were scaled on a three-point Likert scale as well. Motivation was measured as interaction and competition of peers. TAM was eventually checked for polarization. The like–dislike camp was also searched for the gender gap. Overall, the expectation was that the results would be encouraging for achieving good learning outcomes; however, when the TAM factor records constant positive values, a sort of complacency could, in the end, hamper the learning outcomes. For instance, if attitude (6 out of 10 subjects) is inversely associated with motivation (4 out of 10 subjects), additional factors should be checked as well (see Table 1). Female students were prone to exhibit lower attitude rate toward technology use in comparison with male participants, yet the values were still positive for both groups. This preference was extended to in-person education when subjects were asked to relate it to online education.

### 4.3. Perceived Behavioral Control

As informed by social cognitive theory, perceived behavioral control refers to individuals as agents of their own actions, emotions, and goals. Therefore, this section insisted on the self-reflexive behavior of individuals. It looked for proof that self-agency is the driving force that makes individuals take benefits of online schooling. The polar peer's survey method was also used, and a number of factors were scaled in order to determine participants' behavior. For this article, the following factors were selected: ease of use (difficulties to obtain access to online education because of skills), complexity (in comparison with in-person education), facilitating conditions (sharing room and equipment with siblings), and accessibility (having a device and enjoying mobility due to Internet outlets and availability).

Three-point Likert scales were used, for which 3 = more, 2 = same as before, and 1 = less (difficult). Students were asked to score the ease of use, perceived complexity, sharing room and equipment with colleagues and siblings (factors to describe Kemp et al.'s (2019) facilitating conditions) as well as the degree of accessibility. Skills were considered as a sort of new bike, as students have used computers since childhood. Home and office were usually mentioned as Internet stations for most of the online schooling. Malls, cafés, and other public places were just occasionally used. Outlets availability and networks density eventually decide the level of accessibility.

Distribution of perceived abilities to perform online education tasks (self-efficacy) under stress (i.e., public emergency) and home isolation are presented in Table 2.

### 4.4. Cognitive Engagement

Students were asked whether they had more or less to learn or to attain grades online in comparison with in-person education. They had to choose on a three-point scale where 3 = more, 2 = same as before, 1 = less. The extended group of factors includes class assignments, feedback from professors, examinations, and grades. The answers are checked for whether they expose a standard dispersion and, as such, to look for a consistent group of factors involved in online education. These factors are at the core of education as they represent the profound involvement and individual participation in education in general. Except for time-consuming, a factor which is not directly related to this group, cognitive engagement factors present weak deviation from a normal distribution. At first sight, combining these factors does not result in a significant change or difference for online education in comparison with in-person students' engagement. Yet, the $\chi^2$ value for each factor should be weighed against an increase in students' cognitive engagement.

Except feedback from professors, the chi-square test for factors that measure cognitive engagement is closer to the critical value in comparison with any other group in this research. For example, the test value for the feedback from professors (which is highest in the cognitive engagement group) seems to be quite the same as before, while time-consuming chi-square test is found to increase by three times as rated in the self-efficacy group (see Table 2). This research documents that students need up to three times more time

to complete homework and to achieve the same grades with online education; however, the more time they spend online, the less capacity they have to keep focus.

Therefore, relevant factors suggest that expectation or desire for efficacy of online education to be the same as in-person education had been unrealistic. The chi-square test is closer to the critical value for motivation, homework overload, and abandon studies, yet all factors are expected to pass the null hypothesis test.

### 4.5. Social Interactivity

Half of the respondents mentioned absence of human interaction as a challenge during the COVID-19 lockdown. When asked what they missed most, 6 out of 10 students indicated colleagues and friends in both quantitative and qualitative measurements. At first sight, learner–instructor interaction presented certain importance only for 2 out of 10 students (feedback from professors, see Table 2) while impersonal teaching was reported by 1 out of 10 subjects, that is, 10 percent of students exposed to online education reported impersonal teaching. Taking into account that motivation is the product of competition and cooperation with peers as conducted by the instructor, 5 out 10 students reported increased homework load, while 9 out 10 students complained about the extra time they needed for it. Similar perception was reported for increased class assignments. After all, it is important having a real teacher Tichavsky et al. (2015). Important educational load was transferred to students during the COVID-19 lockdown, and individual work compensated social distancing. Exams and grades were evaluated as being less important by 5 students out of 10. Circumstantial factors associated with learner–learner interactivity, such as missing study trips and open air activities, were also reported by 24 and 31 students, respectively (see Table 3).

### 4.6. Qualitative Data

In most of the EU countries, online education came up as a contingency. This research included a number of open questions that looked for specific attitudes with regard to online education. Participants were offered the opportunity to express their personal experience. The answers were grouped into themes (educational fields) that reflect both positive and negative experiences. Table 3 displays a number of them that are relevant to this article. There were situational and environmental challenges (circumstantial) encountered, both educational and emotional (Aguilera-Hermida 2020).

First, self-efficacy revealed a number of situational challenges indeed. Students exhibited a number of negative attitudes related to worries about the pandemic, missing colleagues and friends, and spending too much time in their own room. With online schooling, a circumstantial social field had just developed. It exhibited specific social characteristics as most of the education activities moved online. Students were home yet they were busy most of the time. One student in the University of Bucharest stated that "the pandemic stole two years of my life" as everyone had to adapt to a new everyday lifestyle. It was also "tiresome and time-consuming".

Second, students reported a number of educational challenges, such as difficulties focusing and impersonal teaching, that they were exposed to. They always used the sound function of their device, but not the camera as well. For some it was a good thing as "I did not attend classes before, as I was anxious and shy, so online was better and my relationship with professors had improved". For others, it was the other way around because "I didn't like that it was impersonal, and I was away from colleagues and professors". That is a sort of student anonymization that has occurred with the online education.

Third, participants eventually mentioned anxiety and lack of motivation as emotional challenges. One felt disoriented and lost when not being in touch with colleagues and without proper feedback from professors. One student noted down that "the line between workspace and relaxation space disappeared". Traditional private and public fields had shrunk, while the computer-mediated relationship expanded to a great extent. The logic of connective action (Bennett and Segerberg 2012) not only extended to the online schooling

but absorbed it. Taking into account the role of emotions in everyday life (Jasper 1998), it seems appropriate to look for the substitute socialization that the digital pretends to offer to traditional education as well. Regarding emotional challenges during the COVID-19 lockdown, "students reported stress, anxiety, being worried about getting sick (COVID-19), and changes in their mental health" (Aguilera-Hermida 2020, p. 5). However, it is not so much about mental health as defining the new emotional normal, which is associated with online socialization and that looks for increasing interactions online to compensate for the diminished face-to-face relationships. In this study, participants reported personal improvements due to online education, such as better computer skills and increased abilities to interact with professors and peers (see Table 3).

Fourth, a number of clear positive outcomes came out of this research. An improved sense of safety was reported by a good number of participants with both physical (home) and emotional (family) dimensions. Let us remember that the public orders that imposed the lockdown were intended to improve public safety. Participants shared with family their worries as well their hopes of fighting the COVID-19 pandemic. As such, one student mentioned that "at the beginning I felt as in a permanent vacation, being able to stay all day with my family, and I felt safe from the virus". On the other hand, curfews kept them inside longer than expected. However, for some of them "I loved that it was comfortable, to sit in my room in pajamas and follow classes with my coffee beside me".

Last, but not least, new learning was achieved. They relate to performing multiple activities online and using new digital applications. Half of the students reported they operated at least two applications while being online (education and either job, social media, or games). Computer skills improved for up to 5 out of 10 students who took the courses online. As such, a student from the University of Bucharest mentioned that "I liked that I had so much time, and I could do so many activities and take care of myself. I liked that I learned to use the technology better".

## 5. Interpretation of Results

Results point out that e-learners and e-teachers enjoyed decent (facilitating) conditions and good accessibility (Internet outlets and speed). Technology use approval rate is 8 out of 10 while attitude toward use records an approval–disapproval rate of 6 to 4 against 1 neutral (see Table 1). Accessibility score points out that 8 students out of 10 are satisfied with it, while the other 2 cases are related to poor social conditions and less to educational technology. Worth mentioning here is that universities offered equipment (tablets) to students in need. In specific conditions, but for a rather limited number of students, permission was given for accommodation within the campuses during the lockdowns.

Self-efficacy under stress presents good scores as well, to the extent that anxiety reported by 5 students out of 10 (see Table 2) is balanced by safety and family comfort reported by 80% of the subjects interviewed (see Table 3). In the qualitative measurement, participants also reported personal improvements due to online education, such as better computer skills and increased abilities to interact online with professors and peers.

In detail, this research points out that both professors and students improved their capacity to use computers for online education. A number of 46 students reported clear improvements in computer skills. Compared with the ratio of technology acceptance and attitude toward use (48 and 60, respectively; see Table 1) it proves that the ratio of students with lower technology skills who had to move online was 4 out of 10. This ratio confirms the assumption that students with lower computer skills exhibited a lower perception of self-efficacy (individual trust with the technology) and therefore had a lower cognitive engagement (capacity to stay focused) as well.

These results confirm that online education increased students' expectation and positive attitude towards online teaching technology. During the COVID-19 lockdown, education technology provided a necessary substitute, and learners, instructors, and family took the opportunity to keep education running.

The question now is how social distancing converts interaction with peers and instructors to such an extent that it impacts basic educational factors such as motivation and satisfaction with content. Taxonomies of education technologies, such as that provided by Kemp et al. (2019), underline the prevailing roles played by attitude, affect, and motivation in making sense of techno-tools in general. During the lockdowns, the frequencies that measured motivation reveal that 4 out of 10 students reported good motivation while another 4 reported the same motivation level as before moving online.

With a chi-square test value of 20.68, motivation ranks fourth (after accessibility, facilitating conditions, and difficulty) among the factors defining online education for this sample (see Table 1). However, motivation level is not endorsed if compared with affect (the feelings of joy or hate associated with content, as described by Triandis 1980, p. 211; quoted in Kemp et al. 2019, p. 2400) and with cognitive engagement (the ability to focus). That is, students report a good level of motivation but they exhibit a rather modest capacity to stay focused during the lengthy online sessions and to receive meaningful content in this time as well. In numbers, 7 out of 10 students mentioned difficulties focusing, while 8 students out of 10 reported diminished satisfaction with content; however, this is not educational content in itself, but the delivery method (computer-mediated) and the diminished social interactivity due to social distancing.

With the qualitative data (Table 3), subjects also reported difficulties focusing, missing human interaction, and facing impersonal teaching. Yet, in facing the challenge of adaptation, a number of subjects (14 out of 100) reported improvements in having online interaction with their class. They were selected mostly from a group of students that reported poor computer skills at the beginning of the lockdown. With the qualitative data as well, 38 students confirmed that they spent too much time online, while 51 of them reported a very busy schedule in general. Secondary factors that came into view with the qualitative data are related to study trips (internships) and openair activities that were also missed by some 30% of the interviewed subjects.

At this stage, results indicate that two thirds of students did not report difficulties in keeping in tune with the online education process, and they adapted their daily life to the new education delivery method. Up to one third reported diminished motivation alongside less satisfaction with content.

This research looked for additional explanations related to the motivation. As such, self-efficacy was correlated with stress and it resulted that 5 out of 10 students experienced anxiety while 7 out of 10 were worried about the pandemic (see Table 2). The emotions of the public emergency made their imprint on education, as 2 out of 10 students mentioned they were considering somehow delaying or even abandoning studies and returning later on to the college. Social distancing (missing colleagues and friends) somehow amplified anxiety, yet it is difficult to quantify its load.

The cognitive engagement factors measured homework increase by 48% and supplementary class assignments by 35% as well. By all means, online education proved to be time-consuming for 9 out of 10 students. A good part of the educational load was transferred to homework during the COVID-19 lockdown, and individual worktime increased to compensate social distancing. As mentioned in Section 4.5, half of the subjects rated exams and grades as being less important. Circumstantial factors associated with learner–learner non-learning interactivity (games, parties, travel, visits, etc.) were also reported as missing by 24 and 31 students, respectively (see Table 3).

Results also support the second assumption of the research hypothesis and point out that social distancing is able to convert interaction with peers and instructors to such an extent that it impacts basic educational factors, for instance, motivation and satisfaction with content. Up to 30% of the students selected for this research reported diminished motivation and contracted satisfaction with content.

There might be other educational factors to be interrogated, but the six ones selected for this research and particularly three of them—technology acceptance, attitudes, perception, and motivation, and social interactivity—prove to be illustrative of the difference between

online and in-person education, to measure relevant educational dimensions, and to assess social distance's influence on higher education in general.

## 6. Statement and Limitations

This research was conducted in two leading universities that engaged autonomous online education during the pandemic. They exhibited no coordination of policies yet their outcomes are not at variance. The Aguilera-Hermida (2020) approach inspired this research, except for the measurement of the perceived abilities to perform online education tasks that looked for stress's negative influence instead of neutral position. However, a number of limitations should be taken into consideration. The sub-samples have relevance for the limited numbers of students in the departments they were selected from. Concerning the findings, they were presented in a plain format not designed to look more sophisticated than they actually are. In the end, the interpretation of results is valid for the population of students that the research sample was selected from. It is delivered with the expectation that it is instructive for a larger audience. This research received no funds and is free from conflicts of interest.

## 7. Conclusions and Recommendation

This article introduces the main findings of an exploratory research intended to describe the experience of online learning during lockdowns. It suggests that comprehensive research on online education contingency episodes should be engaged at the earliest convenience. International experience should also be assimilated with the intention to improve higher education at home. Comparisons between countries are useful, and transferring good practices helps everyone.

The main hypothesis is that social distancing during the COVID-19 lockdown affected educational factors and this is confirmed for up to one third of students taking part in online education. The interruption of social interaction with peers and professors authenticates the increasing role that technology plays in education. However, new connective technologies are meant to offer not a substitute but a complement for face-to-face interaction. The article selected six educational factors out of some 61 measurement constructs recommended by recent taxonomy of education technology. By correlating factors of technology acceptance with affect and motivation for a sample of one hundred students extracted from two universities, it resulted that online education proliferated positive attitude towards online teaching technology among students during the COVID-19 lockdown. Students exposed good motivation but, on the other hand, they also exhibited a rather limited capacity to stay focused and to receive meaningful content, too.

Therefore, this article recommends for online education to improve and adapt its content and delivery methods to the educational goals that universities usually have. The online experience achieved so far is useful to redesign the role of in-person education as well.

In the end, universities are expected to learn the lessons of the online education and to improve their programs, to allocate resources for digitalization with the intent to augment both online and in-person education.

**Funding:** This research received no external funding.

**Institutional Review Board Statement:** The study was conducted in accordance with the research activities deployed within the University of Bucharest and it is completely in agreement with the European Charter for Researchers and a Code of Conduct for the Recruitment of Researchers (IRB # SAS 562-2022).

**Informed Consent Statement:** Informed consent was obtained from all subjects involved in the study.

**Data Availability Statement:** Not applicable.

**Conflicts of Interest:** The author declares no conflict of interest.

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
