# Peer review of "Social Distancing Impact on Higher Education during COVID-19 Lockdown"

_socsci, doi:10.3390/socsci11090419_

Round 1
Reviewer 1 Report
The research reports results that are congruent with other contemporary literature: The need to define a new social norm online, reduced cognitive engagement, and the need to improve online teaching methods. For this reason the research is useful and worthy to be published, however, I recommend attention to the following items to strengthen the manuscript.
General notes
1. While perfect English is not required, a language editor to clarify passages would help the authors get their message across more clearly, and assist the reader to understand quickly.
2. Please include more citations, particularly when claims are made, such as ‘previous research suggests that…’ etc. I found that many statements could be supported by citations in the literature.
3. Methods would benefit from more clarity with regard to statistics. Specifically, please explain why the Chi-squared tests are used, and which Chi-squared result is listed in tables 1 and 2. I think it is the Chi-squred test for independence, but that is not clear. Methods could be broken up into two clear sections: Data collection, and Data analysis.
4. Lastly, the taxonomy from Kemp et al has seven primary taxonomic categories that are purported to relate to online learning. While not all need to be addressed in the research, it would be better to mention why only a few were selected for focus in this paper. In addition, the category ‘instructional attributes’ would be an obvious category to specifically include in this research, since it includes content, feedback and interaction amongst peers as sub-categories, all of which feature in the results. The authors have the opportunity to specifically reference this taxonomic category in their interpretation of results, or to refactor the paper to include it in section 2.1.
Some specific notes
Introduction
Lines 49-50: “However, the online experience is also relevant to redesign the role of in-person education that is still at the core of higher education in general.”
I’d suggest adding why this is important. For example, how does online learning support, or complement face to face learning? It would be good to provide a reference to back this up. Then, finish with a statement on what this paper adds so that online and face to face modes can successfully work together.
Research design and method
2.1 Selecting the factors
Lines 69-71: “Yet, most of the online education during the Covid-19 pandemics had no ARS to measure instant reaction and therefore to evaluate students’ participation and even less their contribution.
Is there a reference for this claim? Is this just for Europe or certain areas in Europe? How do the authors know?
Lines 72-73: “More often than not teachers addressed live cameras and microphones without acknowledging how many persons online were altogether over ‘there’.”
Same as above. I suggest moving this sentence after the one above in the same paragraph, and starting the new paragraph with “ Research that looks for…”
2.1.1 Technology Acceptance
I would recommend a rethink of this section (ie remove it), as it isn’t deep enough to cover technology acceptance, but also the paragraph isn’t really about technology acceptance. To improve structure and readability, I would suggest a restructure:
Move the first sentence in lines 84-87 up into section ‘2.1 Selecting the factors’, after “More objective measurement asks for factors to be grouped into primary, secondary and tertiary taxonomic groups and to be coupled with appropriate behavioral theories underpinned with technology acceptance and technology use models (Kemp, Palmer, & Strelan, 2019).” then continue with “In research operationalization… to online education.”
At the end of that paragraph, at line 82, I’d recommend adding a statement as to why this taxonomy was used, or was a suitable framework, for this research, followed by a short statement something like ‘with this framework we briefly explore three of the taxonomy’s categories below’. (PS why only three categories? Can the authors address why all seven of the primary taxonomic categories weren’t used? They don’t have to be, but I was left wondering why they didn’t.)
The rest of paragraph 2.1.1, from line 87 to 94, would be better in the introduction, to set the scene and context. With these adjustments, paragraph 2.1.1 can be omitted altogether and I think the ideas flow better.
2.1.2 Attitudes, perception and motivation
There are some claims here that require references. At the least, could the authors include references for the following?
“Attitudes towards use (TAM) refer an individual’s positive, negative or neutral use of the online schooling”. – I think this is poorly worded, as attitude doesn’t refer to use, but a user’s perception of using that technology.
“Most of the subjects had online education since high school. Some others did it earlier. It is part of their daily life. Yet this was the first time someone asked them about attitudes and motivation”. – how do we know this?
“Outside the digital, previous research suggests there is negative correlation between negative emotions and cognitive processes.” – this is correct, but which research? A citation is definitely needed here.
“Good learning outcomes are usually associated with positive attitudes and affectionate emotions.” – citation needed.
The sentence “Hence, what is real cognitive process…to be determined” can be moved to the end of the paragraph. The idea is good, but for readability I’d recommend something like: ‘In order to get comprehensive results, the students in this study have been asked about both traditional and online learning with the aim of gaining deeper insight into their current attitudes.’ It is up to the authors to determine if this fits what they want to say.
“Motivation is driving the subject’s determination to learn, that is students’ wishes to 109 perform.” – citation needed
2.1.3 Perceived behavioural control
“Self-efficacy comprises the subject’s judgments of his or her own capacity to attain specific tasks that peers usually perform.” – correct but a citation is needed.
“Self-efficacy is part of the Social Cognitive Theory as developed by the 1970s.” – citation definitely needed.
“Currently, social cognitive theory considers individuals as self-motivating agents in the engagement of their own actions, emotions, cognition and motivation.” – citation needed
2.1.4 Cognitive engagement
This sentence “The flow of information should be properly proportioned in order to fit the receiver capacities. Usually, adapting information flow to students’ average capacity gives them the opportunity for cognitive engagement.” is really nice. This meaning of flow is different to that described by Kemp et al (which referred to the flow of time, losing a sense of time), but, the authors here use the concept of flow of information in ways that are acceptable to the cognitive abilities student. Really nice concept. Thanks for including.
2.2 Selecting subjects
I’m not sure what this means: “The factors that drive a student’s attitude towards education technology as described by Kemp, Palmer, & Strelan (2019) presented clear features of a social polarization within the larger society.” – could the authors please re-word or explain? It links to line 176: ” taking into account the polarization premises”. Could the authors explain what this polarization pertains to a bit more clearly?
Line 180: I think this should be: “Subsequently, a Pearson Chi-square test of independence was performed to check for answers’ mutual exclusiveness.” If so, could the authors add it, if not, can it be clearer about which Chi-squared test was used?
This sentence ” The Chi-square goodness of fit test is also relevant to check the hypothesis whether expected results of online education are similar to the observed ones of in-person education.” Could be re-worded to make clearer what the Chi-squared null hypothesis is, for example: “The Chi-squared goodness of fit test was used to check the null hypothesis that the results of online education are similar to the results of in-person education. A rejection of the null hypothesis (Chi-squared value greater than the critical value) would indicate that student perceptions of online education are different to those of face to face education.” Also, I didn’t see the results of the Chi-squared goodness of fit test, can they be included in the results?
Author Response
Please find attached my response to reviewer's comments and recommendations

Reviewer 2 Report
Significant changes are required.
Based on the article received, I feel that the manuscript could be reconsidered for publication after considering all the major revisions attached below:
1. Please revise your title. Now it is in two lines. Please make it in a single line.
2. The abstract needs to be revised substantially. There is no indication of results and recommendations in the abstract.
3. Please add COVID-19 and social distancing to the keywords part.
4. Covid-19 should be replaced by COVID-19 in all places in the manuscript.
5. In the Introduction section, please clarify the justification for this research. The problem statement is not well organized and not clear to the reader. The introduction section is too short, and there is no methodological explanation or reasons for choosing the perception index. Also, there is a considerable gap between the novelty of research and the discussion of existing studies. Please make it as elaborate as you can. Substantial changes and revisions are required in the introduction section. Please make a story and try to find out the importance of this research.
6. Please add research questions, hypotheses, and research objectives.
7. In the Materials and Methods section, please give an overall conceptual framework considering methodological aspects. Please revise this section according to your objectives.
8. The justification for selecting a model based on Aguilera-Hermida (2020) is not properly addressed.
9. I am wondering if the sample size is too small (100). Only 50 respondents from one university cannot represent the entire online education system's reflection.
10. Please revise the conclusion section. In the conclusion section, remind the readers about your objective at the beginning of the conclusion section and conclude your result in light of it. Please add the conclusion section carefully and make a recommendation based on your research findings.
11. Please mention the limitations of this study and provide what could be done in the future in the conclusion section.
12. This paper does not follow the social science, MDPI format, or style. Please revise the formatting in the revised version.
Finally, I would like to say that the research topic is crucial during this global pandemic time. Research ideas are also good, but you should focus on the consistency of your writings based on findings.
Author Response

(The authors gave the same response as above.)

Round 2
Reviewer 2 Report
Thank you so much for your great effort to address all comments. I appreciate your hard work. Congratulations on your great work. Wish you all the best.